# Stereoscopic Visual Perceptual Learning in Seniors

**DOI:** 10.3390/geriatrics6030094

**Published:** 2021-09-18

**Authors:** Sabine Erbes, Georg Michelson

**Affiliations:** 1Department of Ophthalmology, Schwabachanlage 6, Friedrich-Alexander-Universität Erlangen-Nürnberg, 91054 Erlangen, Germany; 2Interdisciplinary Center of Ophthalmic Preventive Medicine and Imaging, Department of Ophthalmology, Friedrich-Alexander-Universität Erlangen-Nürnberg, 91054 Erlangen, Germany; Georg.Michelson@uk-erlangen.de

**Keywords:** binocular disparity, oldest old/highest aged patients, ageing, visual perceptual learning, visual performance, stereopsis, learning effect, targeted visual stimulation, neuroplasticity/cortical plasticity

## Abstract

Background: We showed that seniors can improve their stereoscopic ability (stereoacuity) and corresponding reaction time with repetitive training and, furthermore, that these improvements through training are still present even after a longer period of time without training. Methods: Eleven seniors (average age: 85.90 years) trained twice a week for six weeks with dynamic stereoscopic perception training using a vision training apparatus (c-Digital Vision Trainer^®^). Stereoscopic training was performed in 12 training session (*n* = 3072) of visual tasks. The task was to identify and select one of four figures (stereoscopic stimuli) that was of a different disparity using a controller. The tests included a dynamic training (showing rotating balls) and a static test (showing plates without movement). Before and after training, the stereoacuity and the corresponding reaction times were identified with the static stereotest in order to determine the individual training success. The changes in respect to reaction time of stereoscopic stimuli with decreasing disparity were calculated. Results: After 6 weeks of training, reaction time improved in the median from 936 arcsec to 511 arcsec. Stereoscopic vision improved from 138 arcsec to 69 arcsec, which is an improvement of two levels of difficulty. After 6 months without training, the improvement, achieved by training, remained stable. Conclusions: In older people, visual training leads to a significant, long-lasting improvement in stereoscopic vision and the corresponding reaction time in seniors. This indicates cortical plasticity even in old age.

## 1. Introduction

Visual experiences—for example, in the form of repetitive visual stimuli—can improve visual performance without altering the optical apparatus itself [1,2,3,4]. This process is referred to as visual perceptual learning (VPL) which is based on the ability of the brain to re-organize itself and adapt in general, which is referred to as cortical plasticity or neuroplasticity. Based on this knowledge, numerous new computer-assisted methods have been developed to exploit the potential of the brain’s adaptability in a targeted manner [3]. The meta-analysis of 24 studies showed that the use of these new methods has not only led to a verifiable improvement in visual ability but also to a deeper understanding in the field of cortical plasticity [5]. The study “Two-stage model in perceptual learning: toward a unified theory” reported “that an understanding of the mechanisms of VPL is vital to the development and application of clinical interventions” [6] (p. 9 pdf files of ref. [6]). Whereas it was thought years ago that the ability of the brain to re-organize was limited to childhood and adolescence, today we know that this ability is still present until adulthood. Re-learning motor functions after a stroke or amputation are just two examples [7]. Recent studies in the field of visual perceptual learning show that success seems to be independent of age. Even adults can significantly strengthen their visual abilities, despite the age-related decline in vision. Visual motor training sessions show lasting success in young athletes and adults with amblyopia [1,5,8]. The aim of this study was to examine whether there is also a significant improvement in stereoscopic capability and corresponding reaction time in seniors of advanced age. In the field of visual perceptual learning, there are no available studies on groups of test persons over 75 years of age.

## 2. Materials and Methods

### 2.1. Patients and Definitions

Eleven seniors (4 men and 7 women) aged between 79 and 96 years, living in the senior citizens’ home “SeniorenstiftRathsberg” in Erlangen, participated in the study. Initially, 20 subjects who wanted to participate in the study were subjected to an initial test consisting of evaluations of their:Visual acuity with best auto-refraction;Stereo-optic performance/level of binocular disparity (stereoacuity in arcsec) [9], both in static and dynamic stereo tests;Corresponding reaction time (in msec) using vision training apparatus (c-Digital Vision Trainer^®^).

Three-dimensional stimuli with a minimum of 368 arcsec were detected by the remaining 11 participants. For each participant, the stereo acuity just detected in the entrance test was considered as the starting point (threshold), and the reaction time (threshold recognition time) was compared in the following measurements. The threshold value was defined as the value of the smallest seen resolution in terms of stereo vision. 

The training was performed with dynamic stimuli. Tests with static stimuli were performed to measure the success of the training four times during the study (at the beginning, after three weeks, after six weeks, and after six months). A large range of disparity levels was covered. We tested and trained stereoscopic stimuli with 23–368 arcsec, each representing a multiple of the base disparity (23 arcsec). Compared to the athletes in a comparable study [4], age-related restrictions had to be taken into account here, and an initial overview had to be provided. At the end of the training, all participants received a questionnaire in which they were asked to subjectively assess their improvements in stereoscopic ability in everyday situations.

The stereoscopic stimulus becomes visible when looking straight forward in a polarized TV-screen (3D-4K) (Figure 1).

### 2.2. Vision Training 

Twice a week, the seniors had to carry out repeated visual motor training sessions recognizing visual tasks (*n* = 256) on the digital vision trainer for 20 min, each over a period of six weeks. The stereo-optic performance was analysed in terms of the rate of correct decisions for each disparity level presented, as well as the corresponding reaction time: in other words, how quickly the participants made the correct choices. Participants were advised to focus more on the accuracy of the answers than on speed

The vision training apparatus (c-Digital Vision Trainer^®^) presents stimuli consisting of four objects, four rotating footballs, moving towards the test persons (Figure 2). The representation as rotating footballs is done to increase alertness and interest. The plane of the four objects has a fixed stereoscopic disparity to the background. Thus, the objects seem to hover in front of the background. One of the four objects of the stereoscopic stimulus is shown with a disparity difference to the other three objects. Only with regular stereoscopic vision is the stimulus detectable. The disparity difference of the one object to the three other objects differed from 23 arcsec, over 46, 69, 92, 138, 184 arcsec to 368 arcsec. The position of the one object with the disparity difference varied by chance within the arrangement of the four objects. The task of each subject is to mark with a controller the position of the one object which seems to be nearer in the arrangement of the four objects.

The visual tasks were repeated several times, varying the disparity difference of the one object to the others and the position within the arrangement of the stereoscopic stimulus. The reaction time to detect the correct ball and the correctness of the answers were documented. The training objective was to reach the next more difficult disparity level and to improve the reaction time. In the training, each participant was presented with 4 individually adapted levels. Most of the steps were recognized, but there were also previously unrecognized steps. Each of these levels was offered 64 times during the training, which makes 256 runs per training. The order of the offered levels was random. The test persons received feedback after each presented stimulus. Training was carried out exclusively with the dynamic stereo test.

### 2.3. Statistical Analysis

The evaluation of each participant’s stereoscopic vision was based on the static test. In the static stereo test, the rotating balls were replaced by static, monochrome grey discs that did not seem to move towards the test persons. 

The minimal recognized disparity difference in the entrance test was recorded as the individual threshold point of recognition and was, therefore, the starting point for each participant. The reaction time of this distance in the initial test was compared in the subsequent tests. The threshold value was defined as the value of the smallest seen resolution in terms of stereo vision. A selection of seven levels of stereo disparities was made, covering each level of difficulty, to determine the individual’s optimal training range. The results of the initial test, the results after three weeks, and the results after six weeks were analysed. After a further six months without training, an additional final test was conducted. In the tests, all seven levels of the initial test were offered. Thus, improvements outside the trained area could also be recorded. Both the reaction time and the accuracy of ball selection in the different difficulty levels (stereo-acuity/arcsec) were measured by computer. The statistical analysis was performed with SPSS Statistics V 27 and V 28 (1.0.0.-118).

On the one hand, all participants were analysed as a group. To assess whether the training led to a significant improvement in reaction time and stereoacuity, the t-test with paired samples was applied with the aim obtaining a *p*-value less than 0.05. The mean values of the entire group were compared from the starting point to the time after three weeks of training, after six weeks, and six months without training. On the other hand, the intraindividual comparison of each test person was made.

## 3. Results

The evaluation of the success of each training period was carried out with the static test. The training was carried out with the dynamic version. The reaction time increased with increasing levels of difficulty. The effect was measured after three and six weeks of training and after six months without training. We found improvements in stereo acuity in terms of the threshold point of recognition and corresponding reaction times.

During the entrance test, nine volunteers were not able to detect any level of stereo disparity offered to them. Therefore, they no longer possessed stereoscopic vision, or it was not detectable with the methods used. These subjects were excluded from the study.

After six weeks of training in 8 out of 11 subjects, the stereo acuity threshold improved (Figure 3).

A significant improvement in stereoscopic vision was demonstrated with a median improvement of two levels of disparity/stereo acuity.

The stereoscopic vision improved from 138 arcsec (median) to 69 arcsec (median). In 8 of 11 subjects, there was a significant improvement after six weeks, from 148.45 arcsec (mean) to 73.18 arcsec (mean). The improvement was also demonstrated in seven out of ten subjects after six months (69 arcsec (median) and 79.45 arcsec (mean)) (Table 1).

In 7of 11 subjects, we found an improvement in reaction time after six weeks of training (Figure 4 and Figure 5).

The data showed an improvement in reaction time from 936 msec (median) to 511 msec (median) and from 2659.14 msec (mean) to 1076.86 msec (mean). In eight of ten subjects, we found a significant improvement in reaction time after 6 months without training (581.50 msec (median) and 768.41 msec (mean), *p* < 0.05) (Table 2) (Figure 6, Figure 7 and Figure 8).

In addition to the static non-trained test, the same seven levels were offered in the dynamic test for comparison (Figure 9).

## 4. Discussion

We found that visual stereoscopic perception training produced significant improvements in stereo acuity and median reaction time. A long-term improvement has also been demonstrated. Even six months after the training, the subjects were still at the same level as after six weeks. Unfortunately, one participant died during the data collection phase, so that no sustainable long-term improvement could be tested in that individual. 

These findings complement the evidence of training-induced improvements in visual function from comparable studies using the digital vision trainer [2,3,4]. Subsequently, the question arose whether these findings could also be applied to improvements in stereopsis in people with visual impairments, such as abnormal binocular vision [8] and amblyopia [1,5], and people with abnormal binocular vision [10]. Here, possible benefits could be suggested. These clinical successes led to further studies on the mechanisms of VPL, which in turn are essential for the development of newer clinical interventions [6]. In particular, an improvement of stereopsis performance could be detected by analysing reaction times and stereo acuity. In this study, we also analysed these items to detect VPL in seniors as well.

The improvement in stereoacuity was significant in all three tests. The ability to customize difficulty levels provides an extensive data set. Some of the disparity thresholds collected before training were very large (368 arcsec). The goal was first to reach the next more difficult level (improvement by one difficulty level = 23 arcsec). Then, during training, improvement beyond this goal was quickly identified, and a level adjustment was made. Participant number 10 shows an improvement from 368 arcsec initially to 69 arcsec, which corresponds to an improvement of 13 difficulty levels. Such an improvement was not expected at baseline. That such an improvement occurred could be due to the training design. Three already recognized levels and one not recognized level were offered, so that the unrecognized level was influenced by the others. Referring to the study of Godinez, we assume “that performance on a condition is influenced by the previous condition” [11] (p. 10). “A key feature of cue scaffolding is that improvements made in the previous condition potentially influence the depth error of the condition that follows” [11] (p. 9). Participants were instructed to focus more on correct ball selection than on speed of selection. Nevertheless, improvements in reaction times could also been demonstrated. 

The data suggest that the dynamic test improved reaction time more than the static test procedure (Table 3 and Table 4). On the one hand, the experience effect could have been the decisive factor; on the other hand, the texture and movement of the balls in dynamic training increase alertness.

Recording and analyzing reaction times is a useful method to identify specific and non-specific effects of an intervention. In the literature, there are several works in various scientific fields dealing with reaction times [12,13,14,15,16,17]. Although the questions may be different, reaction times seem to be a suitable method to determine progress. Our measurements include response times for different levels of difficulty. Before and after training, an increase in reaction times can be observed with increasing difficulty of stereo acuity (Figure 10 and Figure 11).

The literature also shows an increase in reaction times for increasingly difficult visual tasks. More difficult tasks are associated with a higher cognitive performance [13,14,15,18]. As an example, the slowing of reaction time represents the typical Stroop effect, where reaction time increases due to processing conflicts, because unfamiliar actions require greater attention [14]. In Kaltner’s study [13,18] on mental rotation, longer reaction times when recognising one’s own body images indicate an increased demand on cognitive resources.

The item reaction time used allows direct conclusions to be drawn about information processing speed and can be regarded as a cognitive determinant. Based on the findings of Salthouse (Salthouse’s Processing Speed Theory), information speed is the key component in the link between age and cognition [18,19]. General reaction times reflect different numbers of process steps in information processing (in our study: sensory final encoding, comparison, and motor response) depending on the task; an improvement in reaction time suggests an optimized sequence in one of these processes. Developmental and age-related influences manifest themselves clearly in this variable, as studies on interindividual comparison show: for example, when comparing people of different age groups or when comparing athletes and non-athletes [2,3,4]. Differences in performance, whether between children, seniors, and adults or between athletes and non-athletes, are expressed in differences in the increase in reaction times with increasing angular disparity [18]. In Kaltner’s study (2015), an increase in angular disparity meant a higher level of difficulty; in our study, a low angular disparity meant a higher level of difficulty and a concomitant increase in reaction time [13,18]. 

By comparing reaction times before and after visual training, conclusions can therefore be drawn about the improvement in perceptual speed, which correlates with the values of fluid intelligence and short-term memory [18]. The improved reaction times and stereo acuity levels therefore allow a direct conclusion to be drawn about an improvement in processes at higher cognitive levels, in the sense of visual perceptual learning [6].

Studies with seniors are rare, showing longer reaction times in older people [13,16]. These can be attributed to different circumstances: a decrease in visual acuity, decrease in muscle strength and mobility, and developmental and age-related degeneration in certain areas of the brain. In Kaltner’s study, both children and seniors showed slower reaction times [13,18]. According to the authors, developmental and age-related slowdowns in processing speed should always be examined against the background of sensory and motor performance. Despite losses in these two variables, seniors were able to improve, which again indicates an existing learning-induced plasticity in the brain regardless of motor and sensory limitations. This success is mainly due to the cutting-edge methods used, which use virtual reality to produce learning-induced neuroplasticity for rehabilitation and enhancement in binocular disparity performance in advanced age, as called for in Wang’s study [20].

The most progress was made in participants with the worst stereoacuity; this finding adds to results of Godinez, where “participants with worse initial stereoacuity thresholds show a higher PPR, i.e., greater improvement” [11] (p. 8). Due to their advanced age, the subjects suffered from numerous ophthalmological impairments such as cataract surgery, macular degeneration, uveitis, and keratitis. Participant Number Six, for example (age: 96), had a visual acuity of 0.4 in the right eye and 0.5 in the left eye in addition to suffering from macular degeneration, hyperopia, astigmatism, presbyopia, and sicca syndrome. Nonetheless, he showed the greatest improvement in reaction time. The success in training seems to be independent of the optical apparatus; only visual perceptual learning seems to be decisive. Patients with severe eye problems usually show a tendency to retreat and thereby expose themselves to less optical stimulation, which influences the stereo vision/depth perception. Targeted stimulation, therefore, shows the enormous potential of the brain to adapt to visual perceptual learning. The better the visual acuity, the better it was from the outset and the lower the chances for improvement. Even with significant loss of visual acuity, visual perceptual learning (VPL) can be used to achieve a 100% increase in reaction time. The methods used in this experimental study are a good example of targeted optimised training paradigms using virtual reality that can lead to learning-induced neuromodulation [19].

This study indicates that the visual stereoscopic performance of seniors can be sustainably improved with the help of visual perceptual learning. In general, most studies do not include people in the age group of this study. However, demographic change shows us that it is precisely this age group that should become the focus of interest in medical interventions. The demonstrated improvements in reaction time suggest that seniors still have an enormous potential for re-organization in the brain, or cortical plasticity. Cortical plasticity can also be a benefit in old age and is a prerequisite for many types of therapies and preventive measures.

A limiting factor of the study is the small number of subjects (11 people). A second limitation is the selective attrition of study participants, which limits the generalizability of the findings. The exclusion criteria used in the initial test should be reconsidered. Due to the surprisingly good results, which went far beyond expectations, it can be assumed that even for people who initially did not recognize any level, a sustainable effect could be achieved through the training. Another limitation is that it cannot be assumed that the participants have constant vision over the entire test period. Due to progressive eye diseases (sicca syndrome, macular degeneration), vision can be affected. An attempt was made to ensure constant conditions throughout the entire test period (same room, same lighting conditions, anamnesis before each training session) and to keep external influences as low as possible (no participant underwent acute ophthalmological treatment during the test period, no planned eye operations).

## 5. Conclusions

Our promising results provide abasis for further studies: for example, a randomised controlled trial that examines the effectiveness of the programme with a comparison group. Elderly patients should not be excluded from newly developed therapy concepts in the field of visual perceptual learning due to cognitive or physical limitations. Particularly in the field of rehabilitation or dementia prophylaxis, there are many therapeutic and preventive approaches, similar to astudy in stroke patients [7].

## Figures and Tables

**Figure 1 geriatrics-06-00094-f001:**
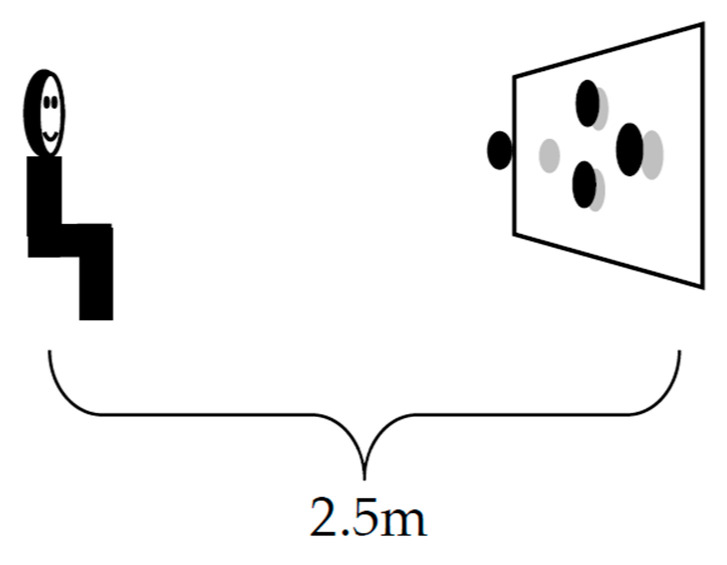
Schematic drawing of the experiment setup: Test person sits centrally in front of the screen looking at the centre of the screen. The distance between participants and test screen was 2.5 m. All tests were presented on the same polarized 3D-TV with 4K (Philips 32PFL5008K/81ck, 32 Zoll, Full HD, 3D with a resolution of 3840 × 2160 dpi).

**Figure 2 geriatrics-06-00094-f002:**
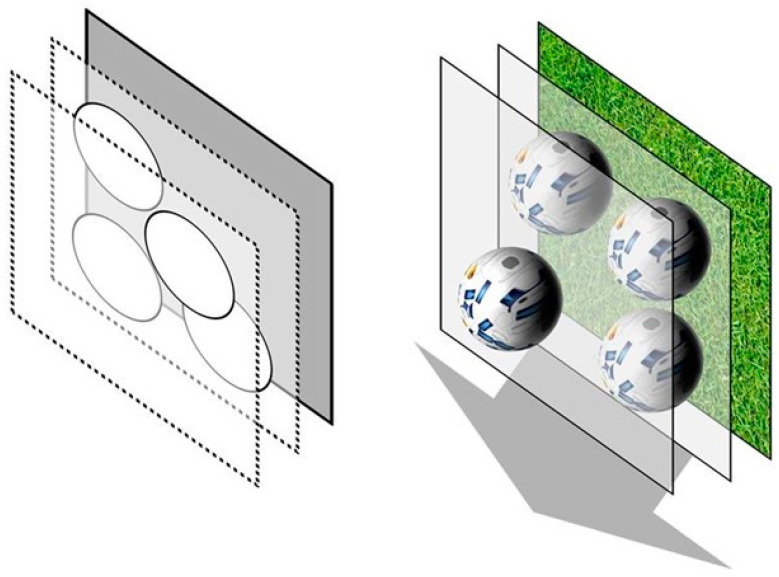
Illustration of the static stereo test (**left**) and dynamic stereo test (**right**). The target objects of the static test are stationary, while the target objects of the dynamic test are constantly moving towards the observer [3]. (Reproduced with permission from Georg Michelson, Extended stereopsis evaluation of professional and amateur soccer players and subjects without soccer background; published by *Front. Psychol. Mov. Sci. Sport Psychol*, 2014.)

**Figure 3 geriatrics-06-00094-f003:**
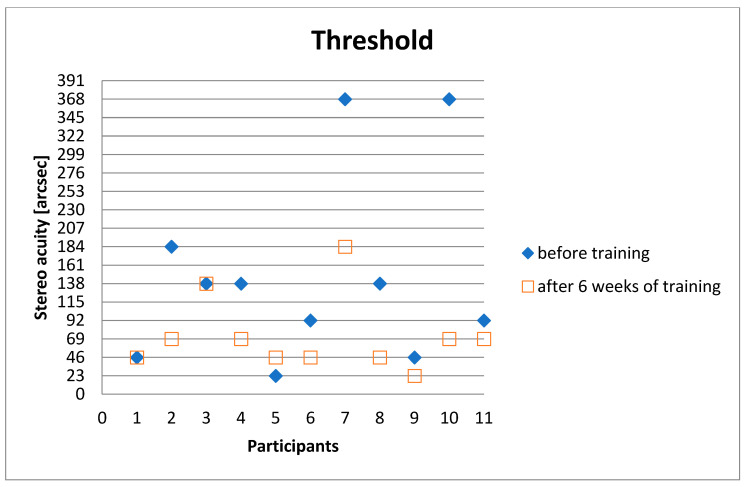
Stereo acuity thresholdof the static stereo test before training and after 6 weeks of training from all 11participants (11). Twenty-three arcseconds represents the most difficult level to detect and 368 arcsec the easiest level to detect. Each line corresponds to a disparity level and a different degree of difficulty.

**Figure 4 geriatrics-06-00094-f004:**
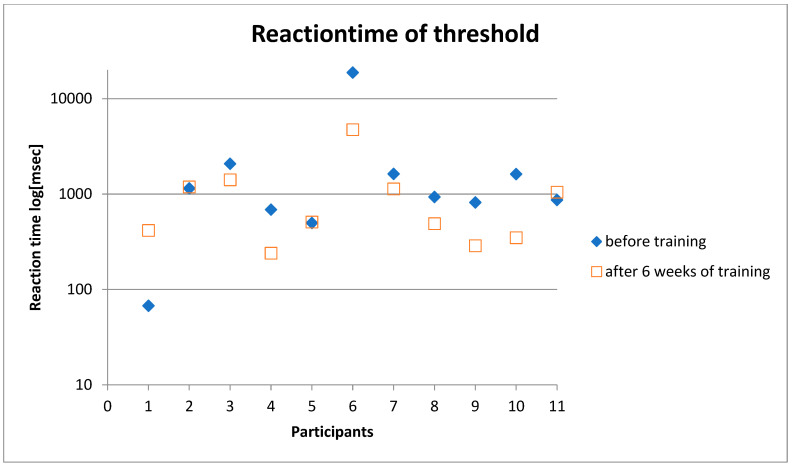
Reaction time recorded by the static test before training and after 6 weeks of training of every participant. Participant number six shows an improvement of 14.131 msec, so the reaction times were presented as a logarithmic scale in milliseconds.

**Figure 5 geriatrics-06-00094-f005:**
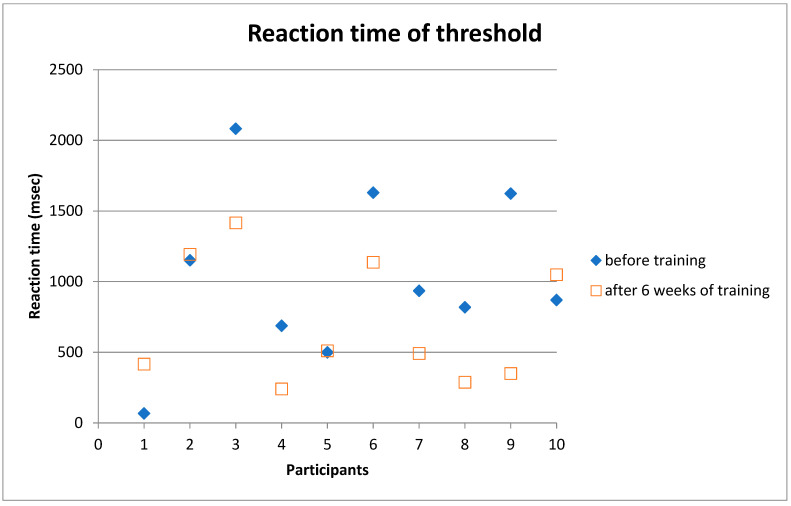
Reaction time recorded by the static test before training and after six weeks of training of ten participants without the participant showing an improvement of 14.131 msec.

**Figure 6 geriatrics-06-00094-f006:**
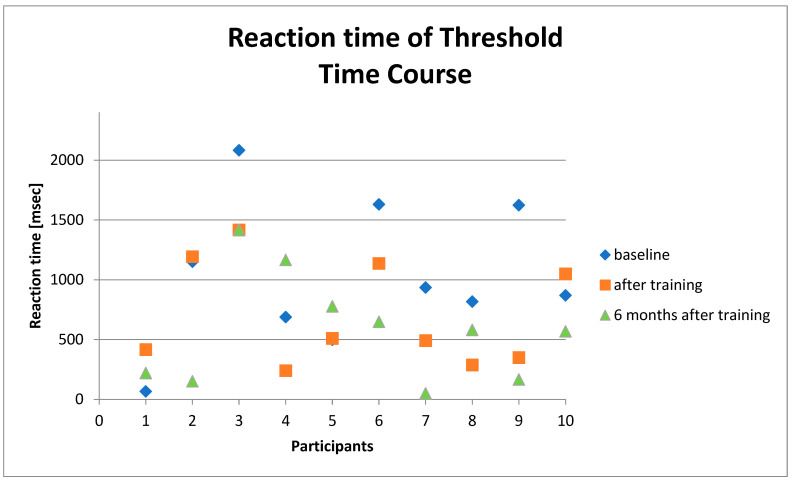
Individual time course of reaction time before training, after training, and after six months without training of 10 participants.

**Figure 7 geriatrics-06-00094-f007:**
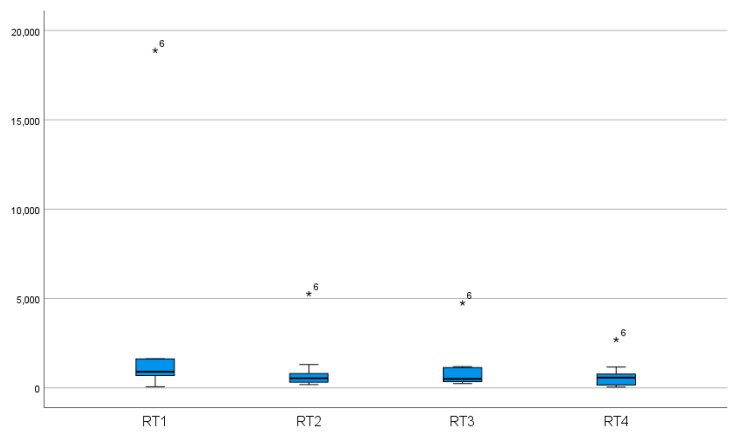
Reaction time (msec) of the individual threshold stereo acuities for the static stereo test. The results of the total group before training (RT1), after three weeks of training/*n* = 1536 trials (RT2), after six weeks of training/*n* = 3072 trials (RT3), and after six months without training (RT4). *, 6: Participant number six shows an improvement of 14.131 msec and a much slower reaction time at the beginning of the training than the rest of the group and represents an “extreme outlier” in the box plot diagram.

**Figure 8 geriatrics-06-00094-f008:**
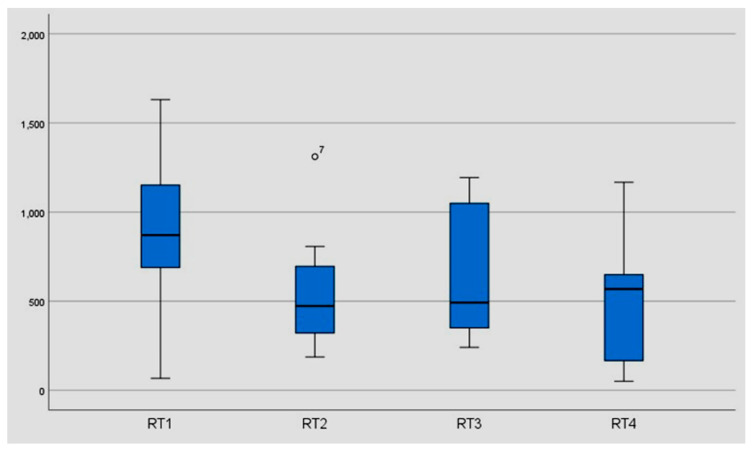
Reaction time of the individual threshold stereo acuities for the static stereo test. The results of the total group before training (RT1), after three weeks of training/*n* = 1536 trials (RT2), after six weeks of training/*n* = 3072 trials (RT3), and after six months without training (RT4). For better illustration, presented without the results of participant number six. ◯, 7: Participant number seven represents a “mild outlier” in the box plot diagram.

**Figure 9 geriatrics-06-00094-f009:**
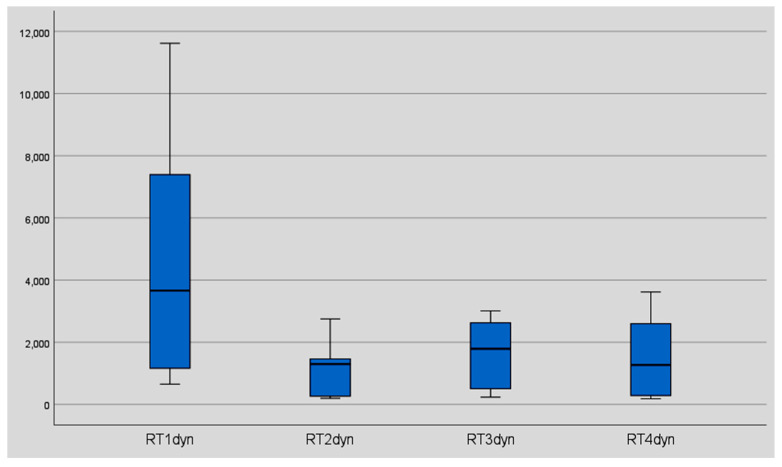
Reaction time of the individual threshold stereo acuities for the dynamic stereo test. The results before training (RT1dyn), after three weeks of training/*n* = 1536 trials (RT2dyn), after six weeks of training/*n* = 3072 trials (RT3dyn), and after six months without training (RT4dyn) of the total group.

**Figure 10 geriatrics-06-00094-f010:**
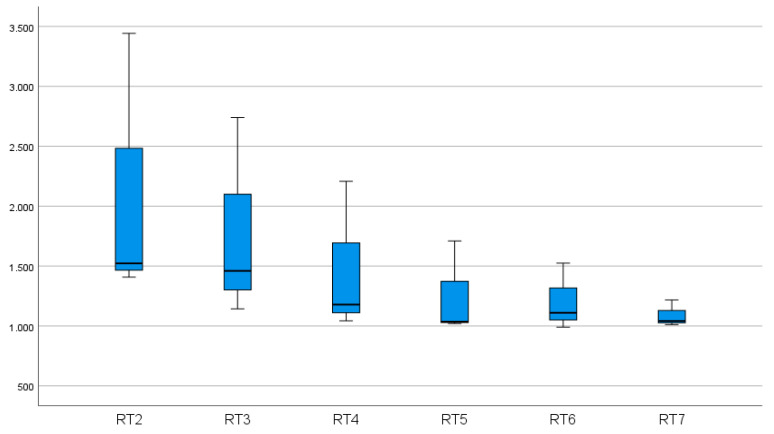
The sum of the reaction times of 10 participants as a function of the stereo acuity level after the training. Forty-sixarcseconds (RT2/level 2) represented the most difficult level and 368 arcsec (RT7/level 16) the easiest level.

**Figure 11 geriatrics-06-00094-f011:**
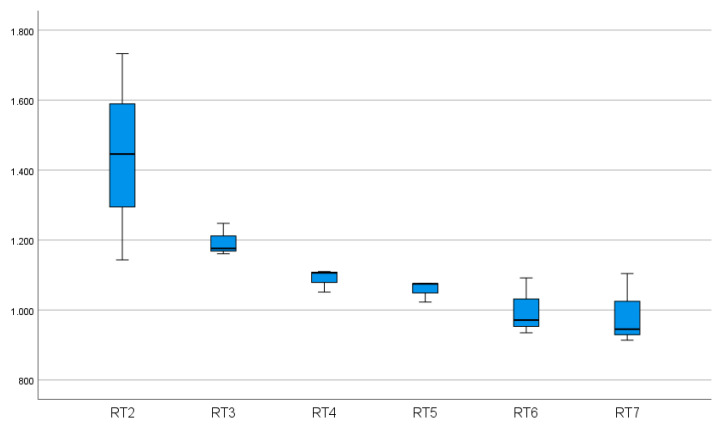
The sum of the reaction times of 10 participants as a function of the stereo acuity level before training. Forty-sixarcseconds (RT2/level 2) represented the most difficult level and 368 arcsec (RT7/level 7) the easiest level.

**Table 1 geriatrics-06-00094-t001:** Development of the stereo acuity in arcsec. Development of the stereo acuity in arcsec of the individual threshold before training and after six weeks of training, listing the difference in arcsec as well as the logarithmic of the difference.

Threshold Stereo Acuity				
Stereo Acuity	Before Training	After 6 Weeks	Difference	Log
subject #1	46	46	0	0
subject #2	184	69	115	0.43
subject #3	138	138	0	0
subject #4	138	69	69	0.30
subject #5	23	46	−23	−0.30
subject #6	92	46	46	0.30
subject #7	368	184	184	0.30
subject #8	138	46	92	0.48
subject #9	46	23	23	0.30
subject #10	368	69	299	0.73
subject #11	92	69	23	0.12
median	138	69	/	/
mean value	148.5	73.2	75.3	0.2

**Table 2 geriatrics-06-00094-t002:** Development of the reaction time in msec of the individual threshold before training and after six weeks of training, listing the difference in msec as well as the logarithmic.

Reaction Time	Before Training	After 6 Weeks	Difference	Log
subject #1	67.5	417	−349.5	−0.79
subject #2	1152	1194	−42	−0.02
subject #3	2083	1417	666	0.17
subject #4	688.5	241	447.5	0.46
subject #5	500	511	−11	−0.01
subject #6	18,879	4748	14,131	0.60
subject #7	1631	1137.5	493.5	0.16
subject #8	936	492	444	0.28
subject #9	819	288.5	530.5	0.45
subject #10	1624	350	1274	0.67
subject #11	870.5	1049.5	−179	−0.08
median	936	511	/	0.26
mean value	2659.1	1076.9	1582.3	0.4

**Table 3 geriatrics-06-00094-t003:** Representation of the development of the median and mean value of the entire sample group’s stereoacuity, reaction time, and level of significance in the static test.

		Before Training	After 3 Weeks	After 6 Weeks	After 6 Months without Training
Threshold (arcsec)	Median	138	46	69	69
/	Mean value	148.45	64.28	73.18	79.45
Reactiontime (msec)	Median	936	606.5	511.5	581.5
/	Mean value	2659.14	1127.82	1067.95	768.41
Reaction time *p* value (compared to the initial reactiontimes)	/	/	0.095	0.059	0.048
Threshold *p* value (compared to the initial threshold)	/	/	0.024	0.026	0.045

**Table 4 geriatrics-06-00094-t004:** Representation of the development of the median and mean value of the entire sample group’s stereoacuity, reaction time, and level of significance in the dynamic training.

		Before Training	After 3 Weeks	After 6 Weeks	After 6 Month without Training
Threshold (arcsec)	Median	46	46	46	46
/	Mean value	135.91	46	52.27	50.18
Reactiontime (msec)	Median	2986.5	1367	1700	1700
/	Mean value	4358.68	1509.59	1618.18	1541.05
Reactiontime *p* value (compared to the initial reactiontimes)	/	/	0.03	0.014	0.014

## Data Availability

The data presented in this study are available on request from the corresponding author.

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
