# Peer review of "Stereoscopic Visual Perceptual Learning in Seniors"

_geriatrics, 2021, doi:10.3390/geriatrics6030094_

Round 1

Reviewer 1 Report

This is an interesting study of the vision perception among elderly subjects. However, there are few concerns that need to be addressed. Specific: 1. The abstract structure should be divided into background, methods, results and conclusions. 2. The material and methods sections: please divide it into three subsections i.e. patients and definitions, vision training, statisitical analysis. 3. The material and methods section: line 57 - this number is the result please move into the results section 4. Please provide more details about statistics 5. There is no need to subdivide the results section 6. Discussion section: you should discuss the limitations of the present study 7. Conclusions: lines 370-373 please move to the discussion (the study group is too small to conclude this).

Author Response

Point 1:

The structure of the article has been adapted according to your suggestions: new structure of the abstract, Results section now no longer subdivided, The Material and Methods section is now divided into 3 sections.

Point 2:

I was not quite sure about the line number (line 57), so I moved the suspected section to the Results section.

Point 3:

Regarding statistics, I have added the following section to the methods part of the manuscript:

The statistical analysis was performed with SPSS Statistics V 27 and V 28 (1.0.0.-118). On the one hand, all participants were analyzed as a group. To assess whether the training leads to a significant improvement in reaction time and stereoacuity, the t-test with paired samples was applied with the aim obtaining a p-value less than 0.05. The mean values of the entire group were compared from the starting point to the time after three weeks of training, after six weeks and six months without training. On the other hand, the intraindividual comparison of each test person was made.

Point 4:

Limiting factors: I have added the following paragraph to the section discussion:

A limiting factor of the study is the small number of subjects (11 people). A second limitation is the selective attrition of study participants which limits the generalizability of the findings. The exclusion criteria used in the initial test should be reconsidered. Due to the surprisingly good results, which went far beyond expectations, it can be assumed that even for people who initially did not recognize any level; a sustainable effect could be achieved through the training. Another limitation is that it cannot be assumed that the participants have constant vision over the entire test period. Due to progressive eye diseases (sicca syndrome, macular degeneration), vision can be affected. An attempt was made to ensure constant conditions throughout the entire test period (same room, same lighting conditions, anamnesis before each training session) and to keep external influences as low as possible (no participant underwent acute ophthalmological treatment during the test period, no planned eye operations).

Point 5:

The conclusion section was revised and expanded to include the following statement:

Our promising results provide the base for further studies, for example, a randomised controlled trial that examines the effectiveness of the programme with a comparison group.

Reviewer 2 Report

In the discussion beginning with line 311, Kaltner's study should be referenced at its first mention rather than at the end of the paragraph.  This study is well known to psychologists, but to other readers of Geriatrics, especially in medicine, to my knowledge, it is little known.  And the reference is to a doctoral thesis, which may make it difficult to obtain.  Is there a better reference available?  Line 312 is difficult to understand and perhaps recognition of own appears to have a word missing.  ?one's own  Again for Salthouse, is there a better citation than to Kaltner's thesis?

Line 355, visual acuity is better than visual accuracy

In the paragraph starting at line 346, I suggest several changes.  Line 349 - ophthalmological instead of optical.  Participant is misspelled.  This whole discussion assumes these elderly individuals have stable vision throughout the period of testing, but that is a shaky assumption.  Hyperopia, astigmatism and presbyopia, as well as childhood scarring should not change, but macular degeneration certainly can.  Uveitis and sicca syndrome will vary widely over the period of testing.  Tendency to retreat leading to less optical stimulation is a shaky hypothesis.   If the eyes are open the retinal photoreceptors are constantly stimulated.  Cataract surgery is uncommonly an impairment, but cataract certainly can be.  Surgery is done to improve the impairment and has a very high success rate.

Cortical plasticity is an interesting phenomenon.  As the authors show, old dogs can learn new tricks.

Author Response

Point 1:

In accordance with your suggested changes, the reference to Kaltner's study is already made at the first mention. In addition, source number 13 is now given. This source contains the same theses as in Kaltner's doctoral thesis, except that the doctoral thesis is more detailed and extensive.

Point 2:

For Salthouse, the source has been extended by the number 19.

Point 3:

Section discussion:

The fact that stable vision cannot always be assumed over the entire test period has been included in the Limitations section.

We have tried as best as possible to create stable test conditions (same room, same lighting conditions). Your objections that stable vision cannot be assumed are correct, but it seems all the more astonishing that such a clear improvement could be detected despite these fluctuations, which can hardly be prevented. Therefore, the cognitive performance of the older participants seems to be relatively independent of these fluctuations. Correct, some of the diseases (macular degeneration) can continuously lead to deterioration of vision, interventions such as cataract op in turn lead to improvement.

Point 4:

Hypothesis: Tendency to retreat leads to less visual stimulation

The tendency to retreat leads to less exposure of older people to situations that specifically stimulate depth perception.

The retina is continuously exposed to optical influences in the form of stimulation of the photoreceptors as soon as the eyes are opened, but a targeted stimulation that leads to a requirement of the higher cognitive functions in the sense of neuromodulation (cortical plasticity) is seen, according to our thesis, in the confrontation with more complex optical stimuli. Such a kind of intensive stimulation as in our test series may not exist in everyday life, but young, active people are exposed to more stimulation with regard to stereo vision (sports: handball, football, fencing) Older people who withdraw, for example, also participate less in road traffic.

These withdrawal tendencies could be subjectively confirmed by the participants through the previously conducted conversations with the test persons. Problems in relation to the assessment of road traffic were mentioned. This uncertainty in relation to depth perception can promote the tendency to withdraw.

Round 2

Reviewer 1 Report

I am satisfied with authors' reply